

# Vertical distribution of soil seed bank and the ecological importance of deeply buried seeds in alkaline grasslands

Ágnes Tóth[1,2,3], Balázs Deák[1], Katalin Tóth[1], Réka Kiss[1], Katalin Lukács[1], Zoltán Rádai[1], Laura Godó[1], Sándor Borza[1], András Kelemen[1,2], Tamás Miglécz[4], Zoltán Bátori[2], Tibor József Novák[5] and Orsolya Valkó[1]

[1] Lendület Seed Ecology Research Group, Institute of Ecology and Botany, Centre for Ecological Research, Vácrátót, Hungary
[2] Department of Ecology, University of Szeged, Szeged, Hungary
[3] Doctoral School of Environmental Sciences, University of Szeged, Szeged, Hungary
[4] Hungarian Research Institute for Organic Agriculture, Budapest, Hungary
[5] Department of Landscape Protection and Environmental Geography, University of Debrecen, Debrecen, Hungary

Corresponding author
Orsolya Valkó,
valko.orsolya@ecolres.hu

## ABSTRACT

**Background:** Soil seed banks play a central role in vegetation dynamics and may be an important source of ecological restoration. However, the vast majority of seed bank studies examined only the uppermost soil layers (0–10 cm); hence, our knowledge on the depth distribution of seed bank and the ecological significance of deeply buried seeds is limited. The aim of our study was to examine the fine-scale vertical distribution of soil seed bank to a depth of 80 cm, which is one of the largest studied depth gradients so far. Our model systems were alkaline grasslands in East-Hungary, characterised by harsh environmental conditions, due to Solonetz soil reference group with Vertic horizon. We asked the following questions: (1) How do the seedling density and species richness of soil seed bank change along a vertical gradient and to what depth can germinable seeds be detected? (2) What is the relationship between the depth distribution of the germinable seeds and the species traits?

**Methods:** In each of the five study sites, four soil cores (4 cm diameter) of 80 cm depth were collected with an auger for soil seed bank analysis. Each sample was divided into sixteen 5-cm segments by depth (320 segments in total). Samples were concentrated by washing over sieves and then germinated in an unheated greenhouse. Soil penetration resistance was measured *in situ* next to each core location (0–80 cm depth, 1-cm resolution). We tested the number and species richness of seedlings observed in the soil segments ($N = 320$), using negative binomial generalized linear regression models, in which sampling layer and penetration resistance were the predictor variables. We ran the models for morphological groups (graminoids/forbs), ecological groups (grassland species/weeds) and life-form categories (short-lived/perennial). We also tested whether seed shape index, seed mass, water requirement or salt tolerance of the species influence the vertical distribution of their seed bank.

**Results:** Germinable seed density and species richness in the seed bank decreased with increasing soil depth and penetration resistance. However, we detected nine
germinable seeds of six species even in the deepest soil layer. Forbs, grassland species and short-lived species occurred in large abundance in deep layers, from where graminoids, weeds and perennial species were missing. Round-shaped seeds were more abundant in deeper soil layers compared to elongated ones, but seed mass and ecological indicator values did not influence the vertical seed bank distribution. Our research draws attention to the potential ecological importance of the deeply buried seeds that may be a source of recovery after severe disturbance. As Vertisols cover 335 million hectares worldwide, these findings can be relevant for many regions and ecosystems globally. We highlight the need for similar studies in other soil and habitat types to test whether the presence of deep buried seeds is specific to soils with Vertic characteristics.

## INTRODUCTION

Soil seed banks, including dormant and non-dormant seeds, present on the soil surface and in the soil, play an important role in population dynamics and diversity maintenance in almost all ecosystems throughout the world (*Bakker et al., 1996*; *Douh et al., 2018*; *Sloey & Hester, 2019*). They serve as reservoirs for seeds that have already become independent of their mother plants in terms of their metabolism and are able to germinate or acquire this ability in the future (*Csontos, 2001*). Soil seed banks also play a considerable role in the long-term preservation of genetic variation in plant populations and in shaping spatial and temporal distribution of species, influencing the structure and composition of future plant communities (*Bossuyt & Honnay, 2008*; *Münzbergová et al., 2018*). In addition, they may have a crucial role in the spontaneous recovery and restoration of various habitats (*Ludewig et al., 2021*; *Sloey & Hester, 2019*), as they might contain the seeds of the target habitat even when its species have already disappeared from aboveground vegetation (*Hopfensperger, 2007*; *Shiferaw, Demissew & Bekele, 2018*; *Wang et al., 2020*). As seed banks usually include seeds from the earlier successional stages, they also function as a so-called "successional memory", providing an insight into the processes of past vegetation dynamics (*Bakker et al., 1996*; *Bossuyt & Honnay, 2008*; *Ma et al., 2021*). Recent studies also emphasize their role when identifying and evaluating potential climate-change microrefugia (*Ashcroft et al., 2012*).

Different vegetation types vary in their seed bank characteristics, such as seed density, seed bank species richness and also in similarity level between the species composition of the seed bank and vegetation (*Bakker et al., 1996*; *Bossuyt & Honnay, 2008*; *Hopfensperger, 2007*; *Plue et al., 2020*; *Wang et al., 2020*). Species richness of the seed bank and the similarity between seed bank and vegetation were found to be higher in grasslands than in forests (*Plue et al., 2020*). In general, seed bank density in grasslands is higher than in forests, but lower compared to wetlands (*Bossuyt & Honnay, 2008*; *Hopfensperger, 2007*). It is important to note that there may be major differences in the seed densities of different

grassland types. The diversity and density of a soil seed bank is the highest in the most stressed grassland types, such as continental and seashore salt-affected grasslands. In these types of grasslands, regeneration from the persistent soil seed bank may play an essential role in maintaining the species richness of the aboveground vegetation due to the unfavourable environmental/growing conditions (*Bossuyt & Honnay, 2008*; *Hopfensperger, 2007*; *Plue et al., 2020*; *Valkó et al., 2014*).

Although several classification systems have been set up to classify seed banks into different types, the system proposed by *Thompson, Bakker & Bekker (1997)* became the most widely used one. It defines three seed bank types (*i.e.*, transient, short term persistent and long-term persistent) based on the longevity of seeds in the seed bank. Plant species can be classified into certain types of soil seed bank based on their relative frequency in aboveground vegetation and seed bank, and on the relative prevalence of seeds in the upper and lower soil layers (*Csontos, 2001*). It is a general trend that the more persistent the seeds of a given species, the higher its proportion found in the deeper soil layers (*Bekker et al., 1998*).

A considerable proportion of the seed bank occurs in the layers close to the surface, mostly in the top 5 cm, and its density and the number of germinable seeds decline with depth (*Bekker et al., 1998*; *Csontos, 2001*; *Shiferaw, Demissew & Bekele, 2018*). The vertical distribution of soil seed bank is shaped by several factors, such as soil type, seed size and seed shape based on the results of previous studies focussing on surface layers of the seed bank. For instance, larger seeds are less likely to enter the deeper soil layers by movement down along cracks or burial by soil-dwelling animals than small ones (*Bekker et al., 1998*; *Shiferaw, Demissew & Bekele, 2018*; *Thompson, Green & Jewels, 1994*). Moreover, small seeds tend to have greater longevity (*Bu et al., 2016*), therefore they have a higher chance (*i.e.*, longer time) to preserve their viability while being deeply buried (*Shiferaw, Demissew & Bekele, 2018*; *Thompson, Band & Hodgson, 1993*).

Soil seed bank studies may contribute to management activities that aim at maintaining the species richness of the natural habitats. It is an especially important task, as the area of natural habitats is in constant decline. However, our knowledge on the restoration potential of soil seed bank in most ecosystems is incomplete. The reason for that is that vast majority of studies examined only the uppermost soil layers, usually the upper 10 or 20 cm, as the seed bank typology systems also consider these layers (*Thompson, Bakker & Bekker, 1997*). Hence, our knowledge on the depth distribution of seed banks and the ecological significance of deeply buried seeds is rather limited, nonetheless information on the soil seed bank persistence of characteristic species may help to predict the effectiveness of habitat restoration, especially when it relies to spontaneous vegetation processes (*Ludewig et al., 2021*; *Poschlod, Kiefer & Fischer, 1995*; *Saatkamp et al., 2019*). It would be crucial to know whether the deep soil layers can be suitable propagule sources for the recovery of communities after degradation.

The aim of this study was to examine the soil seed bank to a depth which, to our best knowledge, has not been done so far except for one study in moving sand dunes where soil layers are constantly re-organized (*Qian et al., 2016*). We studied the fine-scale vertical depth distribution, the species composition, and the seedling density of the soil seed bank

to a depth of 80 cm. Our model systems were inland alkaline grasslands which are priority habitats of the EU Natura 2000 network (Pannonic salt steppes and salt marshes, 1530). These habitats were developed under continental climate and harsh environmental conditions. Their soil characteristics can be specified by high electric conductivity, alkalinity and compactness and shrink-swell potential, resulting in the development of seasonally opening and closing deep cracks, which is classified as Vertic Solonetz according the WRB (*IUSS Working Group, 2015*). Harsh environmental conditions induce the formation of persistent seed banks and the high likeliness of crack-formation increases the possibility of deep burial of seeds (*Burmeier et al., 2010*; *Valkó et al., 2014*). In order to explore the potential of deeply buried seed bank in ecological processes and habitat restoration we asked the following questions: (1) How do the seedling density and species richness of soil seed banks change along a vertical gradient and to what depth can germinable seeds be detected in alkaline grasslands? (2) What is the relationship between the depth distribution of the germinable seeds and the species traits?

## MATERIALS AND METHODS

### Study area

The study was conducted in five study sites located in the Hortobágy National Park, East-Hungary. Hortobágy is the largest salt-affected inland grassland area in Europe. The landscape is composed by a mosaic of various plant communities; it is characterized by alkaline grasslands and marshes, loess grasslands with scattered arable fields, floodplains and wooded habitats (*Deák et al., 2014*). The climate is temperate continental. The mean annual temperature is 9.5 °C and the average annual rainfall is 550 mm, with a maximum in June (80 mm). The average temperature and average precipitation show a considerable fluctuation from year-to-year (*Fick & Hijmans, 2017*).

The soil reference group of the study area is classified as Vertic Solonetz (*IUSS Working Group, 2015*). Vertisols and Vertic horizons are heavy clayey soils with a minimum of 30% clay content to a depth of 50 cm or more, characterised by clay minerals having high shrink-swell potential. Despite their compactness, Vertic soil horizons show a notable temporal and spatial variability. Because of the shrink-swell effect of the clay, dry seasons result in the formation of deep desiccation cracks. Due to the shrinking and swelling behaviour, the upper 10–20 cm of these soils has a mixed granular structure. Most of these types of soils are neutral or alkaline, because of their calcareous smectic clay parent material. The average pH values of Vertisols range from 6.0 to 8.0, however higher pH values can occur as well. High electrical conductivity (EC) is a typical feature of Solonetz, caused by the high base saturation and its sodium-dominated composition, together with high exchangeable sodium percentage (ESP) (*Coulombe, Wilding & Dixon, 1996*; *IUSS Working Group, 2015*).

Sampling was carried out in natural stands of dry alkaline grasslands (association: Achilleo setaceae-Festucetum pseudovinae). These short grasslands occur in dry areas and typical on Vertic Solonetz soils (*Deák et al., 2014*). The dominant grass species is *Festuca pseudovina*, typical forb species include *Achillea collina*, *A. setacea*, *Plantago lanceolata* and *Trifolium* spp. (*Deák et al., 2014*).

## Sampling design

Samples were collected after snowmelt, during late February 2019. In each site, four soil cores (4 cm diameter) of 80 cm depth were collected with a soil auger for soil seed bank sampling (a total of 20 soil core samples). The samples were divided into sixteen 5-cm segments by depth: 0–5, 5–10, 10–15, 15–20, 20–25, 25–30, 30–35, 35–40, 40–45, 45–50, 50–55, 55–60, 60–65, 65–70, 70–75 and 75–80 cm, and each segment was placed separately in a plastic bag (320 soil segments in total). Samples were then stored at 5 °C for 3 weeks. Soil penetration resistance was measured *in situ* with Eijkelkamp Penetrologger next to each core sampling location down to 80 cm depth with 1-cm resolution. Permission for the research was granted by the Trans-Tisza Environmental, Nature Protection and Water Inspectorate (approval no 6646/08/2014).

## Seed bank analysis

Soil seed banks were studied with the seedling emergence method. Samples were concentrated by washing them through sieves according to the method of *Ter Heerdt et al. (1996)*. Vegetative parts and fine soil materials were removed using a 3-mm mesh sieve and then using a fine sieve with a mesh width of 0.2 mm. As a result, sample volumes were reduced considerably. The concentrated samples were spread out in a thin layer (3–4 mm) on trays filled with potting soil topped with an approximately 8-cm layer of sterilized potting soil. Control trays were filled with the same sterilized potting soil to filter out possible wind-borne and potting soil contaminants. The trays were randomly placed in an unheated greenhouse, at the Botanical Garden of the University of Debrecen, where germination was carried out from April to November 2019 under natural light conditions and regular watering. Seedlings were counted regularly, identified when possible then removed. Unidentifiable seedlings were transplanted into separate pots and grown until identifiable. At the end of June no new seedlings emerged, so watering was stopped to mimic arid summer conditions that are typical in the studied ecosystem. Dried sample layers were carefully broken through and stirred, and then at the end of August watering was restarted, which was continued until early November. Nomenclature of the species followed *Király (2009)*.

## Classification of species

Plant species were categorized into two morphological groups (graminoids and forbs) and into two ecological groups (grassland species and weeds), using the Hungarian Flora Database (*Horváth et al., 1995*). Species were arranged into simplified life-form groups based on Raunkiaer's life-form categories. These groups were short-lived species (Th, TH) and perennials (H, G, Ch).

Seed length, seed width and seed thickness values were collected for each species from *Schermann (1967)* to calculate seed shape index (*Bekker et al., 1998*). Seed shape was expressed as the variance of three seed dimensions (length, width, and height), after standardizing all three values, so that length was considered as 1 (dividing all three values by the length). As a result of this, seed shape index was independent of seed size. The more round-shaped a seed was, the closer the index value was to 0, while the more elongated was
the seed, the closer the index value was to 1. The average seed mass of each species was collected from the database of *Török et al. (2013)*. Ecological indicator scores for salt tolerance (SB), and relative water requirement (WB) values (*Borhidi, 1995*)–were assigned to the species. Based on these trait data, community weighted means (CWM) of salt tolerance (SB), relative water requirement (WB), seed mass and seed shape index were calculated by weighting with the number of the germinated seedlings in each layer. The classification of each recorded species into morphological, ecological, and life-form groups, as well as their SB, WB, seed mass and seed shape index are given in Appendix 1.

## Statistical analysis

For all data handling and analyses the R statistical software (ver. 4.0.1; *R Core Team, 2021*) was used. We tested the total number of seedlings, and the total number of species observed in the soil segments ($N = 320$), using negative binomial generalized linear regression models (GLMs; utilizing the *glmmTMB* R-package: *Brooks et al., 2017*), in which sampling layer and penetration resistance were predictor variables, with control for their interaction. Although these variables show substantial correlation, re-scaling prior to model fitting (subtracting the mean from all values and dividing by the standard variation), and controlling for their interaction mitigated the adverse effect of their collinearity, as shown by the low variance inflation scores from the fitted models (5> for all variables and models; tested with the "performance" R-package: *Lüdecke et al., 2021*).

Similarly, we fitted negative binomial models on the number of seedlings and germinated species separately for each level of our categorical variables, with sampling layer and penetration resistance as predictors, with control for their interaction as well. To test the effect of sampling layer and penetration resistance on community-weighted means (CWMs) we used log-linked Gamma GLMs, with sampling layer and penetration resistance as predictor terms. Due to the non-significant interactions, we left out interaction terms from the final models on CWMs. In cases when a given variable had a minimum value of zero, we transformed it by incrementing by one in order to be able to fit Gamma models.

In the preliminary analyses we fitted all models with sampling repetition nested in sampling site used as random factor, but due to the low (close to zero) variances we excluded random effects from the final models.

## RESULTS

During the study 402 seedlings germinated, which corresponds to an overall seedling density of 15,870 seedlings/m$^2$. A total of 51 vascular plant species (and one unidentifiable forb seedling) were detected, of which 180 seedlings of 12 graminoid species and 222 seedlings of 39 forb species were identified (Appendix 1). Short-lived species comprised 60% of the total species and 47% of the total number of germinated seeds. Grassland species were present in greater numbers in the soil seed bank than weeds, both in terms of the number of species (37 *vs* 14 species, respectively) and the number of germinable seeds (374 *vs* 27 seeds, respectively).

**Table 1 Results of the generalized linear models (GLMs) about the effect of burial depth on the dependent variables.**

| Dependent variable | Estimate | Standard error | p |
|---|---|---|---|
| No. of seedlings | **−0.5043** | **0.1271** | **<0.001** |
| Species richness | **−0.2505** | **0.0387** | **<0.001** |
| Forbs, no. of seedlings | **−0.4004** | **0.0785** | **<0.001** |
| Forbs, species richness | **−0.2415** | **0.0416** | **<0.001** |
| Graminoids, no. of seedlings | −0.7413 | 1.1923 | 0.535 |
| Graminoids, species richness | **−0.2758** | **0.0782** | **<0.001** |
| Perennials, no. of seedlings | −0.7351 | 0.9855 | 0.456 |
| Perennials, species richness | **−0.3106** | **0.0772** | **<0.001** |
| Short-lived spp., no. of seedlings | **−0.3779** | **0.0749** | **<0.001** |
| Short-lived spp., species richness | **−0.2284** | **0.0424** | **<0.001** |
| Grassland spp., no. of seedlings | **−0.5518** | **0.1917** | **0.004** |
| Grassland spp., species richness | **−0.2589** | **0.0456** | **<0.001** |
| Weeds, no. of seedlings | **−0.2434** | **0.0670** | **<0.001** |
| Weeds, species richness | **−0.2513** | **0.0635** | **<0.001** |
| CWM, seed mass | 0.0055 | 0.0309 | 0.860 |
| CWM, WB | −0.0084 | 0.0074 | 0.259 |
| CWM, SB | −0.0150 | 0.0129 | 0.251 |
| CWM, seed shape index | **−0.0037** | **0.0019** | **0.048** |

**Note:**
Results of the generalized linear models (GLMs) about the effect of burial depth on the seedling number and species richness of the morphological, ecological and life-form groups, as well as on the CWMs of the studied traits. Significant effects are marked with boldface.

The most abundant species in the soil seed bank were *Juncus compressus* (37% of total seedlings), *Trifolium angulatum* (12%), *Gypsophila muralis* (9%), *Inula britannica* (6%), *Matricaria recutita* (4%), *Plantago lanceolata* (3.5%) and *Festuca pseudovina* (2.5%). These seven species were accounted for about 75% of the total germinable seed number. Of the seven most numerous species in seed bank samples, four were short-lived and three were perennials.

The soil seed bank density and the species richness decreased significantly with soil depth (GLMs, $p < 0.001$ in both cases; Table 1, Fig. 1). The largest decline was shown between the first (0–5 cm depth) and second layers (5–10 cm) of the soil. The upper 0–5 cm soil layer contained 75% (302 seedlings) of all germinated seeds, whereas their proportion was 25% (100 seedlings) in the deeper layers (5–80 cm). In the uppermost soil layer (0–5 cm) 12,016 seedlings/m$^2$ seedling density was detected, while in the deepest layer (75–80 cm) this value was 358 seeds/m$^2$. There was only one soil layer, the 60–65 cm depth, from which no germination has occurred. Although, there were 18 species that only germinated from the first soil layer (0–5 cm depth), 34 species had seedlings in the deeper soil layers. The most frequent species from these deeper layers (5–80 cm) were *Juncus compressus* (18 seedlings), *Inula britannica* (15 seedlings), *Gypsophila muralis* (eight seedlings), *Trifolium angulatum* (eight seedlings) and *Amaranthus retroflexus* (five
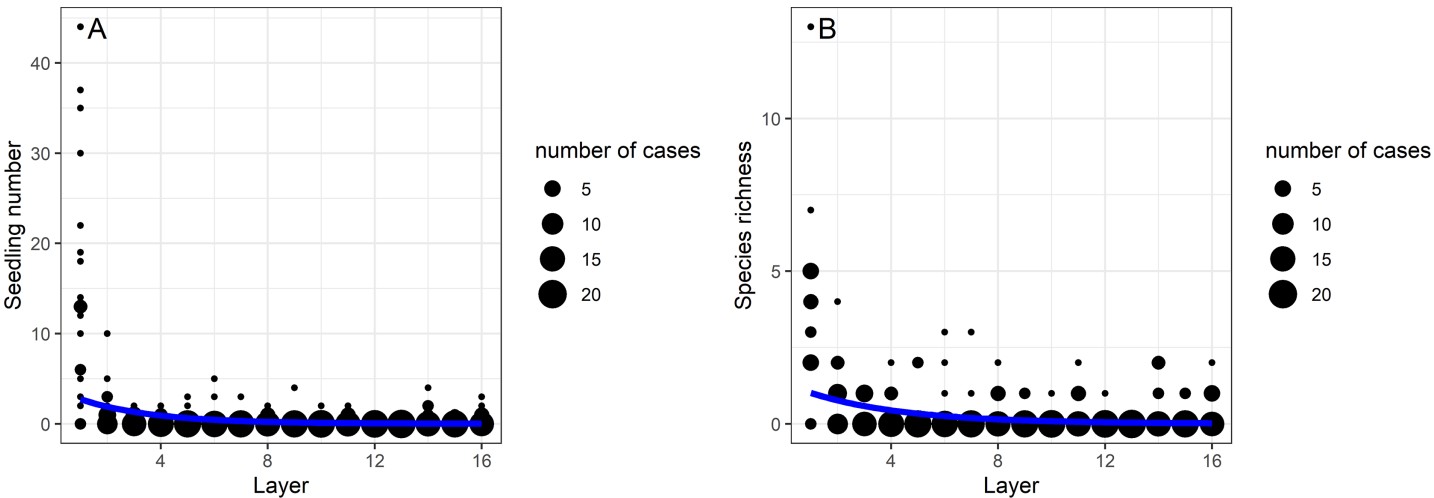

**Figure 1 The effect of burial depth (layers 1–16) on the (A) seedling number and (B) species richness in the studied alkali grasslands.**

seedlings). A total of nine seedlings from six species emerged from the deepest layer (75–80 cm depth), namely *Juncus compressus*, *Cerastium vulgare*, *Trifolium angulatum*, *Trifolium repens*, *Vicia hirsuta* and a forb seedling that died before identification. Of these species, three were short-lived and two were perennials.

Generalized linear models (GLMs) showed that depth had a significant effect on the vertical distribution of morphological (graminoids, forbs), ecological groups (grassland species, weeds) and life-forms (perennials, short-lived species) (Table 1). In the zone between layers 9 to 13 (40–65 cm depth), forbs, grassland species and short-lived species were dominant and occurred in large abundance, while graminoids, weeds and perennial species were missing (Figs. 2A–2F). The abundant species found in these layers were *Cerastium vulgare*, *Gypsophila muralis*, *Inula britannica*, *Stenactis annua*, *Trifolium angulatum*, *T. retusum*, *Vicia angustifolia*.

Seeds in the deeper soil layers had significantly lower seed shape index compared to seeds in the upper soil layers (GLM, $p = 0.049$), which means that round shaped seeds were more abundant in the deeper layers than elongated ones. There was no difference in the depth distribution of the seeds among species with different seed mass and ecological indicator values. GLMs showed no effect of burial depth on the CWMs of seed weight, salt tolerance (SB) and relative water requirement (WB) (Table 1).

Based on the average penetration resistance values of each soil segment, it can be determined that soil penetration resistance increased significantly with depth (Fig. 3). Under the 55–60 cm depth, in the deeper soil layers a slight decrease was shown in the soil penetration resistance. With increasing soil penetration resistance, the seedling number and species richness of all studied morphological, ecological and life-form groups decreased significantly (Table 2).

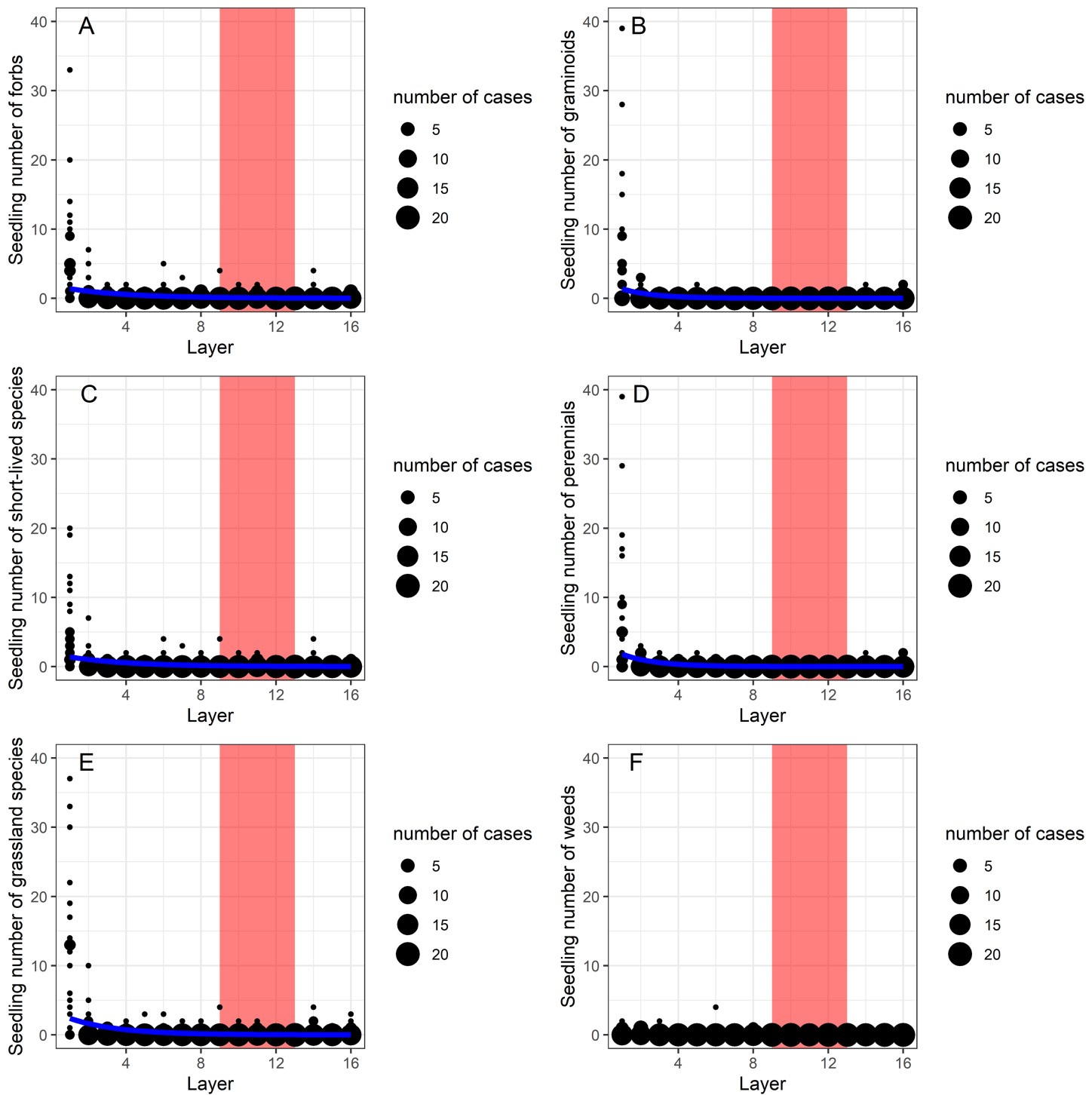

**Figure 2 (A–F) The effect of burial depth on the seedling number of the studied morphological (forbs, graminoids), life-form (short-lived species, perennials) and ecological groups (grassland species, weeds).** The effect of burial depth (layers 1–16) on the seedling number of the studied morphological (forbs, graminoids), life-form (short-lived species, perennials) and ecological groups (grassland species, weeds). Layers 9–13 are marked with red to highlight the contrasting distribution of the different species groups in this zone with high penetration resistance (see Fig. 3).

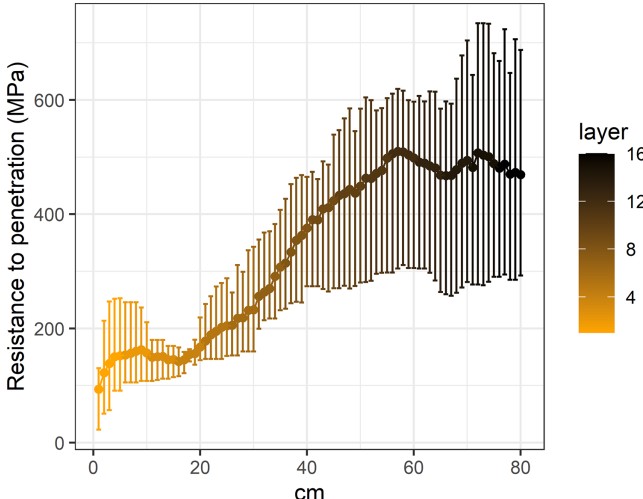

**Figure 3 Penetration resistance curves (0–80 cm) in the studied Vertic Solonetz soils.**

**Table 2 Results of the generalized linear models (GLMs) about the effect of soil penetration resistance on thedependent variables.**

| Dependent variable | Estimate | Standard error | $p$ |
|---|---|---|---|
| No. of seedlings | **−0.0056** | **0.0016** | **<0.001** |
| Species richness | **−0.0036** | **0.0009** | **<0.001** |
| Forbs, no. of seedlings | **−0.0057** | **0.0015** | **<0.001** |
| Forbs, species richness | **−0.0033** | **0.0010** | **<0.001** |
| Graminoids, no. of seedlings | **−0.0055** | **0.0024** | **0.023** |
| Graminoids, species richness | **−0.0044** | **0.0015** | **0.003** |
| Perennials, no. of seedlings | **−0.0061** | **0.0022** | **0.007** |
| Perennials, species richness | **−0.0051** | **0.0014** | **<0.001** |
| Short-lived spp., no. of seedlings | **−0.0053** | **0.0014** | **<0.001** |
| Short-lived spp., species richness | **−0.0030** | **0.0010** | **0.002** |
| Grassland spp., no. of seedlings | **−0.0057** | **0.0017** | **<0.001** |
| Grassland spp., species richness | **−0.0035** | **0.0010** | **<0.001** |
| Weeds, no. of seedlings | **−0.0053** | **0.0018** | **0.004** |
| Weeds, species richness | **−0.0054** | **0.0017** | **0.002** |
| CWM, seed mass | −0.0003 | 0.0008 | 0.754 |
| CWM, WB | −0.0002 | 0.0002 | 0.452 |
| CWM, SB | −0.0004 | 0.0003 | 0.196 |
| CWM, seed shape index | **−0.0001** | **0.0001** | **0.037** |

**Note:**
Results of the generalized linear models (GLMs) about the effect of soil penetration resistance on the seedling number and species richness of the morphological, ecological and life-form groups, as well as on the CWMs of the studied traits. Significant effects are marked with boldface.

## DISCUSSION

### The vertical distribution of seed bank in alkaline grasslands

We found that both the germinable seed density and species richness of the seed bank significantly decreased with soil depth and that seedling density was the highest in the upper soil layer (0–5 cm) in the studied ecosystem. These seed bank patterns are consistent with the results of previous studies analysing the seed bank of different grassland (*e.g.*, *Ma, Zhou & Du, 2010*; *Niknam et al., 2018*), forest (*e.g.*, *Godefroid, Phartyal & Koedam, 2006*; *Jaroszewicz, 2013*) and wetland ecosystems (*e.g.*, *de Souza, Damasceno-Junior & Pott, 2019*). More importantly, despite the sharp decline below the first five centimetres, deeper soil layers also contained germinable seeds (*e.g.*, 1,154 seeds/m$^2$ in the 5–10 cm soil layer) that were able to germinate (if seeds were brought to the surface or near the surface). A novelty of our study is that we found that germinable seeds can be detected in soil layers much deeper than previously noted in other studies.

Based on the reviewed literature and our current knowledge, we believe that this is the second study in which germinable seeds from such a deep soil layer have been detected. The only study that investigated similarly deep burial of seeds was performed by *Qian et al. (2016)*, but they studied the seed bank of mowing sand dunes–which are dynamic systems–where soil layers are constantly re-organized. They found that wind activities and sand burial contribute to the formation of the spatiotemporal pattern of the soil seed banks of active sand dunes. Due to the displacement of the seeds in different directions over time, seed density in active sand dunes showed a relatively constant vertical distribution across the studied sand profile. A significant amount of germinable seeds were found in the deeper soil layers too, which could be a form of adaptation of plant species to the effects of wind erosion on active sand dunes. In addition, their study showed that deeply buried seeds are brought to or near to the surface by the wind, and thus they play an important role in population regeneration and vegetation restoration in active sand dunes.

The majority of seed bank studies carried out in grassland (*Godefroid et al., 2018*; *Jacquemyn et al., 2011*; *Ma, Zhou & Du, 2010*; *Niknam et al., 2018*) and forest ecosystems (*Douh et al., 2018*; *Godefroid, Phartyal & Koedam, 2006*; *Jaroszewicz, 2013*) usually investigated only the uppermost soil layers to a depth of 20 cm. However, the vertical distribution of the seed bank in wetland ecosystems has already been studied at greater depths (*Burmeier et al., 2010*; *Dawson et al., 2019*; *Jauhiainen, 1998*; *McGraw, 1987*; *van der Valk & Davis, 1979*; *Wall & Stevens, 2015*). According to the reviewed literature, these greatest depths were 45 cm (*McGraw, 1987*) and 50 cm (*Burmeier et al., 2010*; *Jauhiainen, 1998*), respectively. This also means that seed bank research has generally neglected the deeply buried seeds. In the present study we found that deeper soil layers (down to 80 cm) may also play an important role in the preservation of at least some seeds of grassland species. Our results showed that the penetration resistance of the soil significantly increased with soil depth. This finding supports our result regarding the vertical distribution of soil seed bank; that is, the more compact the soil was, the fewer seeds it contained.

## Role of plant traits in vertical seed distribution

Forb species were more abundant than graminoids in the seed bank, as has also been shown in other studies (*Douh et al., 2018*; *Jutila, 1996*; *Ma, Zhou & Du, 2010*). Although, short-lived species were dominant over perennials in the terms of species number, the seedling number of perennials was higher, suggesting a higher seed bank density. Seeds of grassland species were represented by a significantly higher density in the soil seed bank than weed species. Although forbs, grassland species and short-lived species occurred in a relatively large abundance in the deeper soil layers, graminoids, weeds and perennial species were scarce. A possible reason for this pattern is that opportunistic germination strategy is vital in the case of short-lived plant species, so they tend to have a greater longevity and might penetrate deeper than perennials (*Jutila, 1996*). The higher abundance of forbs compared to graminoids in the deep soil layers may be due to the fact that many graminoids have elongated seeds, therefore they cannot penetrate as deeply as rounded seeds (see also *Bekker et al., 1998*). Our findings that deeper soil layers were characterised by seeds of grassland species rather than weeds imply that seeds of grassland species have the potential to tolerate harsh environments for a longer time. The aboveground vegetation in the studied habitat type is species-poor and dominated by one perennial grassland specialist grass, *Festuca pseudovina*, with a percentage cover above 60% (see *Deák et al., 2014*; *Valkó et al., 2014*, *2017*). Despite its dominance in the aboveground vegetation, *F. pseudovina* occurred with low density in the soil seed bank. The aboveground vegetation of alkaline grasslands contains very few weed species in low cover, due to the high salt content of the soil (*Valkó et al., 2014*) which was also reflected in the seed bank composition found in the current study.

Our results suggest that there is no difference in the depth distribution of the seeds of species with different seed mass and ecological indicator values. However, we found that seed shape was an important factor in determining the vertical distribution of seeds. Seeds with low seed shape index, *i.e.*, round-shaped seeds, were dominant over elongated ones in deeper soil layers. This result corresponds to the findings of *Bekker et al. (1998)* in relation to seed size, shape and vertical distribution in the soil.

## Implications for conservation and restoration

The periodic drying and rewetting characteristic of soils with Vertic horizon can cause cracks down to a depth of 1 m or more (*Coulombe, Wilding & Dixon, 1996*). This crack-formation may be a major driver of the vertical distribution of the seed bank. This assumption is supported by the findings of *Burmeier et al. (2010)* who studied the impacts of desiccation cracks on soil seed bank formation in a flood-meadow ecosystem. They found that the distribution and composition of the seed bank were different in samples taken along cracks and in samples taken beside them, as these cracks acted as natural seed traps. Desiccation cracks also had an effect on the vertical translocation of the seeds. The survival rate of these entrapped seeds increased with increasing depth, which could imply that in combination with other abiotic factors desiccation cracks cause a selection pressure on plant species to develop long-term persistent seed banks. Furthermore, their study showed that desiccation cracks influenced the establishment of the seed bank,

and therefore, were an important driver of community assembly and dynamics in flood-meadows.

Our findings draw attention to the potential ecological importance of the deeply buried seeds in Vertisols and other reference soil groups with Vertic horizons. Our results originate from a single region and ecosystem type, and the number of samples was also limited by the field and greenhouse capacities. Despite the limitations in the sampling strategy, our results show that germinable seeds in Vertisols occur in deeper layers than previously thought. As Vertisols cover 335 million hectares worldwide (*IUSS Working Group, 2015*) and further extended areas are covered by soils with Vertic horizons, these findings may be relevant for many regions and ecosystems, such as grasslands, savannas, shrublands and open forests, globally. Vertisols occur in the tropic or semiarid regions in Australia, the Americas, India and Northeastern Africa (*Coulombe, Wilding & Dixon, 1996*). To test the generality of our findings, studies are needed in other regions and ecosystems with Vertisols on the vertical distribution of the soil seed bank. We highlight the need for similar studies in other soil and habitat types to test whether the presence of deeply buried seeds is a specific characteristic of Soils with Vertic horizons.

There are different views about the role of the persistent seed banks in the formation of species richness in stressed communities, such as alkaline grasslands (*Valkó et al., 2014*). Some researchers found that clonal spread has a greater role in stressed environment than sexual reproduction, thus, persistent seed bank is less important for habitat restoration and maintaining diversity (*Chang, Jefferies & Carleton, 2001*). In contrast, others showed that despite the temporarily unsuitable environmental conditions, species are able to germinate and establish from the persistent seed bank; therefore, persistent seed banks may play an important role in the vegetation dynamics of communities in stressful environments (*Bossuyt & Honnay, 2008*; *Valkó et al., 2014*). Our results support the latter findings and imply that the persistent seed bank in the deeper soil layers of alkaline grasslands can be a promising source for habitat restoration even many years after degradation.

Our knowledge on the depth distribution of seed banks and the ecological significance of deeply buried seeds is still limited. Our current study suggests that deep soil layers might contain more germinable seeds than previously thought. Our results may (1) contribute to the better understanding of the concept of seed bank persistence; (2) provide information about the past vegetation of a given area, about the past land use practices and past abiotic and climatic changes; (3) provide a better knowledge on the present conservation value of the studied area; (4) contribute to the planning of conservation measures as well as habitat restoration design by providing more detailed information on the seed bank dynamics of a given area.

Based on these results, the question arises whether these deeply buried seeds have a chance for germinate. There are several natural and anthropogenic processes by which the deeply buried seeds may return to the soil surface (*Douh et al., 2018*). For instance, the activity of the soil fauna–especially the movements of soil-dwelling rodents–may enable buried seeds to return to the surface (*Reichman, 1979*; *Valkó et al., 2021*). Soil layers are

mixed during building and construction (such as road and railway construction and mining) and agricultural activities (such as ploughing) that can bring deeply buried seeds to the surface. These activities often occur in the studied landscape, for instance *via* the establishment of drainage ditch systems, soil-filling of channels and restoration of landscape scars (*Deák et al., 2015*; *Valkó et al., 2017*). It has been observed in areas affected by such activities that salt-tolerant species (*Suaeda* spp., *Salsola soda*) absent from the vegetation of the actual landscape, reappeared in the restored sites (*Deák et al., 2015*; *Valkó et al., 2017*). These observations also suggest that through the mixing of soil layers, deeply buried seeds may play an active role in vegetation dynamics. Besides the unintended mixing of soil layers, restoration projects might actively activate deeply buried seeds by targeted topsoil removal. Future studies need to clarify whether and how the deeply buried seeds can be used in restoration projects.

## CONCLUSIONS

In this study we found that in alkaline grasslands with Vertic soil horizon, the density and species richness of soil seed bank decreases along a vertical gradient but germinable seeds can be found also in deep soil layers down to 80 cm depth. This is the first study that found germinable seeds in such deep soil layers in non-sandy soil. Our results suggest that it is important to also consider the deep soil layers in seed bank studies and restoration ecological research as these soil layers can also contain germinable seeds of the native species characteristic of the plant community. Our results showed that round-shaped seeds can be found in deeper soil layers than elongated ones, but seed mass, water requirement and salt tolerance of the plant species did not affect the vertical distribution of the seed bank. Based on the results, the most important research directions may be (1) analysing the vertical distribution of soil seed bank in other ecosystem and soil types, (2) testing the role of natural and anthropogenic processes in mixing the soil layers and bringing the deeply buried seeds closer to the soil surface and (3) to study the possibilities for utilizing the deeply buried seeds of native species in ecological restoration projects.

## ACKNOWLEDGEMENTS

The authors are grateful for Ferenc Báthori and Szilvia Radócz for their assistance during fieldwork, and for Csaba Albert Tóth and István Kapocsi for discussions about the topic.

### Funding

The authors were supported by the Hungarian Research, Development and Innovation Office (grant numbers: NKFI FK 124404 (Orsolya Valkó), NKFI KDP C1762731 (Ágnes Tóth), NKFI PD 137632 (Réka Kiss), NKFI FK 135329 (Balázs Deák), NKFI KDP 967901 (Sándor Borza), NKFI K 124796 (Zoltán Bátori)) and the Hungarian Ministry of Human Capacities (grant number: NTP-NFTÖ-21-B-0095 (Laura Godó)). András Kelemen and Zoltán Bátori were supported by the Bolyai János Scholarship of the Hungarian Academy

of Sciences. The funders had no role in study design, data collection and analysis, decision to publish, or preparation of the manuscript.

## Grant Disclosures

The following grant information was disclosed by the authors:

Hungarian Research, Development and Innovation Office: NKFI FK 124404, NKFI KDP C1762731, NKFI PD 137632, NKFI FK 135329, NKFI KDP 967901, NKFI K 124796.

Hungarian Ministry of Human Capacities: NTP-NFTÖ-21-B-0095.

Bolyai János Scholarship of the Hungarian Academy of Sciences.

## Competing Interests

The authors declare that they have no competing interests.

## Author Contributions

- Ágnes Tóth performed the experiments, analyzed the data, prepared figures and/or tables, authored or reviewed drafts of the paper, and approved the final draft.
- Balázs Deák conceived and designed the experiments, performed the experiments, analyzed the data, prepared figures and/or tables, authored or reviewed drafts of the paper, and approved the final draft.
- Katalin Tóth performed the experiments, authored or reviewed drafts of the paper, and approved the final draft.
- Réka Kiss performed the experiments, authored or reviewed drafts of the paper, and approved the final draft.
- Katalin Lukács performed the experiments, authored or reviewed drafts of the paper, and approved the final draft.
- Zoltán Rádai analyzed the data, prepared figures and/or tables, authored or reviewed drafts of the paper, and approved the final draft.
- Laura Godó performed the experiments, authored or reviewed drafts of the paper, and approved the final draft.
- Sándor Borza performed the experiments, authored or reviewed drafts of the paper, and approved the final draft.
- András Kelemen performed the experiments, authored or reviewed drafts of the paper, and approved the final draft.
- Tamás Miglécz performed the experiments, authored or reviewed drafts of the paper, and approved the final draft.
- Zoltán Bátori conceived and designed the experiments, authored or reviewed drafts of the paper, and approved the final draft.
- Tibor József Novák conceived and designed the experiments, authored or reviewed drafts of the paper, and approved the final draft.
- Orsolya Valkó conceived and designed the experiments, performed the experiments, analyzed the data, prepared figures and/or tables, authored or reviewed drafts of the paper, and approved the final draft.
### Field Study Permissions

The following information was supplied relating to field study approvals (*i.e.*, approving body and any reference numbers):

The Trans-Tisza Environmental, Nature Protection and Water Inspectorate approved this study (6646/08/2014).

### Data Availability

The seedling numbers per all species and per layer (layers 1–16) are available in Appendix 1.

### Supplemental Information

Supplemental information for this article can be found online at http://dx.doi.org/10.7717/peerj.13226#supplemental-information.

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
