# Peer review of "Vertical distribution of soil seed bank and the ecological importance of deeply buried seeds in alkaline grasslands"

_PeerJ, doi:10.7717/peerj.13226_

## Round 0.1 · original submission · Major Revisions

Dear authors

I kindly ask you to take into great consideration the indications of the two reviewers and to resubmit an updated version of your manuscript.

Reviewer 1 ·

Basic reporting

I carefully read the ms “Vertical distribution of soil seed bank and the ecological importance of deeply buried seeds”.
The authors analyzed the distribution of germinable seeds along with other seed traits (e.g. morphology, mass, tolerance to salinity…) up to a depth of 80 cm. The ms is interesting, as it addresses the role of seed banks, focusing on deep layers, which provides important insights in the field of biodiversity conservation and restoration.
I have only one issue that is not very clear to me, that is the reference to “seed density” (e.g line 262, and throughout the text, abstract included), which in my opinion generates confusion. Since the authors used the seedling emergence method (line 185), they actually measured the “seedling density”, i.e. the proportion of germinable seeds. This is fine, but I think that it should be specified more clearly, since possible non-germinable (dormant?) seeds may be part of the seed bank, and some seeds, possibly fallen in deep layers, may have died during the experiment. Thus, I suggest to replace “seed density” with something else (“seedling density”? “germinable seed density”?) throughout the text, when necessary.

Tables and figures: please, note that you provided a double legend for each table and figure

Supplementary file: please, provide a legend for the supplementary file. Explain in the legend the meaning for WB and SB, and add the unit of measurement to seed traits (e.g. “Seed length (mm)”, Seed mass (g? mg?)”

Experimental design

The experimental design and data analyses were well performed

Validity of the findings

No comment

Additional comments

Below I list minor points:
Line 66: did you mean “shaping” instead of “shaping of”?
Lines 85-90: this sentence is not clear: The diversity and density of soil seed bank is generally highest in the most stressed grassland types due to unfavourable environmental conditions? Maybe the sentence needs to be split in two parts.
Lines 127-128: does alkali grasslands correspond to a Natura 2000 habitat? Can you add the habitat code for the environments that you have investigated?
Line 152: did you mean “are classified” rather than “classifies”?
Line 156: delete the repetition (see line 154)
Lines 216-218: How did you calculated the CWM for each layer? Specifically, on which value did you weighted each seed trait? On the number of emerged seedlings? Please explain the procedure that you used
Line 248: replace “seed density” with something else, here and throughout the text (as explained in my comment on the basic reporting)
Lines 248-254: I suggest to represent these data in graphs (e.g. pie charts, or bars)
Line 282: the letters A-F are missing in the figure
Line 342-355: Are forbs more abundant or less abundant than grasses in your study system? Is it possible that seeds from forbs were more abundant than seeds from grasses because forbs have a higher cover percentage than grasses in the grassland? While the relative abundance along the depth gradient of grasses, forbs and other categories assigned to the seedlings that emerged during the experiment can be compared (e.g. “forbs were more abundant in the deepest layer, while grasses were more abundant in the upper layers”), absolute abundance of given categories (e.g. forbs vs grasses, or grassland species vs weeds) should be discussed in relation to the species composition and species cover of the grassland. Please rephrase the paragraph eliminating or addressing comparison of absolute abundances.
Line 369-375: summarize this paragraph avoiding repetitions (see line 133)
Line 398: delete the question mark
Line 425: did you mean “in other” instead of “is other”?

Reviewer 2 ·

Basic reporting

The authors analyze depth distribution of viable seeds in deeper soil layers down to 80 cm depth which has been rarely done so far in soil seed bank studies that are usually confined to the upper 10 cm of the soil profile. Seed soil bank analyses were done at five alkali grassland sites with Solonetz soils that are characterized by a strong shrink-swell potential due to high clay content and the regular occurrence of desiccation cracks during periods of drought that may facilitate the penetration of seeds to deeper soil layers.
The major finding is that viable seeds – albeit in very low densities - may be found down to a depth of 80 cm which has rarely been shown before. The authors argue that this is probably caused by the physical peculiarities of the studied Solonetz soils, in particular due to the regular occurrence of desiccation cracks. In general, the paper is well written, except some cases where phrasing and wording could be improved (see examples Additional comments).

Experimental design

Technically the soil seed bank analysis is well executed with remarkable depth resolution in steps of 5 cm.

Validity of the findings

The major problem I see with this study is the rather limited explanatory power due to the low number of replicates (five) and the confinement to one soil type which is constraining the detection of patterns in the functional characteristics of seeds in deep soil layers and does not a allow a comparison between soil types with and without desiccation cracks to assess their importance the allocation of seeds to deeper soil layers. Therefore, I feel that the results are overall of limited value and validity. Increasing the number of replicates and including another soil type without desiccation cracks could considerably increase the validity of the study. Nevertheless, the finding of viable seeds in such deep soil layers alone is quite remarkable and may probably as such justify publication.

Additional comments

Specific comments

L 48-49: “Forbs, grassland species and short-lived species occurred in large abundance in deep layers, from where graminoids, weeds and perennial species were missing.” I feel the characterization of plant groups here is a bit confusing, due to overlapping characteristics.

L104-106: There should “to” in front of “enter” and “along” in front of “cracks”

L110-111: “As the area of natural habitats is still rapidly declining, it is an important task to maintain their species richness, to which soil seed bank studies may contribute.” There is something wrong with the logic of this sentence.

L113: Skip the first usually in this line.

L 117: “the chances of the effectiveness” sounds a bit weird and redundant.

L 119 skip “usually neglected” (this was said before)

L310-311: You use the verb “found” three times in a relatively short senence.

References: I feel one important reference is missing: Burmeier, S., Eckstein, R.L., Otte, A. & Donath, T.W., 2010: Desiccation cracks act as natural seed traps in flood-meadow systems. - Plant & Soil 333: 351-364.

Fig 1 and 2: Due to the large number of overlapping points (in particular zero values), box plots would probably give a better impression about the distribution of seeds.

---

## Round 0.2 · accepted · Accept

Dear Authors,

All required changes have been correctly addressed and therefore the manuscript can be accepted.

Reviewer 1 ·

Basic reporting

ok

Experimental design

ok

Validity of the findings

ok

Additional comments

I received the revised version of the ms. The points risen during the revision process were properly addressed, so I think that the ms is now ready for publication.